# Usefulness of Imaging Techniques in the Diagnosis of Selected Injuries and Lesions of the Canine Tarsus. A Review

**DOI:** 10.3390/ani11061834

**Published:** 2021-06-19

**Authors:** Justyna Abako, Piotr Holak, Joanna Głodek, Yauheni Zhalniarovich

**Affiliations:** Department of Surgery and Radiology with Clinic, Faculty of Veterinary Medicine, University of Warmia and Mazury, 10-719 Olsztyn, Poland; piotr-holak@wp.pl (P.H.); j_glodek@wp.pl (J.G.); eugeniusz.zolnierowicz@uwm.edu.pl (Y.Z.)

**Keywords:** radiography, computed tomography (CT), magnetic resonance imaging (MRI), ultrasonography, tarsal joint, dog

## Abstract

**Simple Summary:**

Tarsal joint lesions are uncommon in dogs but may lead to serious health problems. The most common diseases involving the tarsal joint are osteochondrosis, fractures and injuries of the Achilles tendon. The basis for the diagnosis of lesions in the tarsus is a thorough orthopedic examination, sometimes performed under sedation. Imaging modalities such as radiography, ultrasonography, magnetic resonance imaging or computed tomography can facilitate the detection and assessment of lesions in the canine tarsal joint. The aim of this paper is to characterize and compare the usefulness of imaging techniques available in veterinary medicine for the diagnosis and evaluation of lesions and injuries affecting the tarsal joint in dogs.

**Abstract:**

Tarsus lesions are not common in dogs, but they can cause serious health problem. They can lead to permanent changes in the joint and, in dogs involved in canine sports, to exclusion from training. The most common diseases and injuries involving the tarsal joint are osteochondrosis, fractures and ruptures of the Achilles tendon. These conditions can be diagnosed primarily through accurate orthopedic examination, but even this may be insufficient for performing a proper diagnosis. Imaging modalities such as radiography, ultrasonography, magnetic resonance imaging or computed tomography can facilitate the detection and assessment of lesions in the canine tarsal joint. This review paper briefly presents some characteristics of the above-mentioned imaging techniques, offering a comparison of their utility in the diagnosis of lesions and injuries involving the canine tarsus.

## 1. Introduction

The tarsus is a joint with a complex structure. It consists of seven short bones that comprise four joint rows, which are themselves surrounded by a common capsule and reinforced with numerous ligaments and tendons [1,2]. This joint acts as a shock absorber, but a lack of support and protection from the surrounding soft tissues makes it prone to injury [2]. Additionally, the tarsal joint may be affected by developmental diseases, such as osteochondrosis, which may produce degenerative lesions. Fractures, sprains and shearing injuries of the tarsus are uncommon in dogs. Central tarsal bone fractures are typical of racing greyhounds that run counterclockwise on a racetrack. These injuries are the result of repetitive microcracks, insufficient repair response, or weakness during bone resorption due to extensive bone remodeling [3]. In 64% of cases, they are accompanied by fractures of the remaining tarsal bones [4], and in each case, secondary bone dislocation occurs [5]. Osteochondrosis (OCD) is a disease in which there is a detachment and mineralization of flaps of articular cartilage, which may lead to degenerative changes in the joint. Osteochondrosis has been reported in dogs, horses, pigs and humans [6,7,8,9]. The ankle joint in dogs is rarely affected by OCD lesions, which are usually observed on the trochlear ridges of the talus [10,11,12,13]. Achilles tendon injuries are most often the result of direct force, leading to the tendon’s tearing or complete rupture. These changes are uncommon in dogs and can manifest themselves as mild to severe lameness and an apparently abnormal limb angle in the joint. The basis for the diagnosis of injuries and changes in the tarsus is a thorough orthopedic examination, sometimes performed under sedation. As long as it allows for the localization of lesions, accurate diagnosis is possible only after performing additional examinations. Radiography is the primary imaging technique used in veterinary orthopedics because it is relatively inexpensive, easily accessible, and usually requires sedation rather than general anesthesia [1]. However, in some cases, taking radiographs is insufficient for assessing and locating lesions. If a soft tissue injury is suspected, an ultrasound may be helpful [14]. Advanced imaging modalities such as magnetic resonance imaging (MRI) and computed tomography (CT) are increasingly used in veterinary medicine. The cost of performing CT and MRI scans is decreasing as equipment becomes more widely available [1]. The aim of this report is to characterize and compare the usefulness of imaging techniques available in veterinary medicine for the diagnosis and evaluation of lesions and injuries affecting the tarsal joint in dogs.

## 2. Osteochondrosis

Osteochondritis dissecans, also known in the literature as osteochondrosis, is a developmental disease of the joints and bones, which affects immature, fast-growing large breed dogs [15,16]. Labrador Retrievers and Rottweilers are over-represented (approximately 70% of all breeds) [17,18], but OCD is also frequently noted in Bullmastiffs and Australian Cattle Dogs. [17,19,20]. The tarsus is the third most frequently affected joint, representing 3–9% of overall cases of canine OCD [15,17,21].

Lesions characteristic of OCD are observed in the tarsocrural joint, within the medial and lateral trochlear ridges of the talus [13,22]. According to the available literature, the medial trochlear ridge is affected in about 75–80% of cases [10,15]. OCD of the lateral trochlear ridge represents only 20–25% of cases [11,15,17,23] and is more common in Rottweilers [10,11,15,24]. According to many authors, the most affected area in the medial ridge is its proximal part [17,18,24], although Beale et al. [25] stated that the plantar area of the medial trochlear ridge is the most affected area, accounting for 80% of all medial ridge OCD cases. In the lateral trochlear ridge, the dorsal, dorso-proximal and proximal parts are most affected [10,11,12]. In approximately 35–75% of cases, OCD lesions occur bilaterally [16,17,23,26], but only 30–50% of dogs exhibit bilateral lameness [11,25,27].

OCD diagnostics based on radiographic examination are difficult because of the complexity of the tarsal structure and its overlapping bony elements [10,18,28,29]. Despite a sufficient number of radiographic views, minor changes involving both trochlear condyles may be overlooked. In most cases, it is not possible to determine how much the joint surface is affected by the disease, or even to obtain information about the number, size and location of osteochondral fragments inside the joint [18]. Moreover, viewing the joint from multiple positions is time consuming and usually requires sedation or general anesthesia [28].

According to Morgan et al. [21], OCD of the tarsocrural joint is present when a radiographic examination shows a defect in the trochlear bone outline in any view, the presence of free bone fragments in the joint, and a periosteal reaction of the distal part of the tibia with modeling of the articular surface. Van der Peijl et al. [16] proposed the following criteria for the assessment of OCD lesions in radiographs: location and size of OCD lesions, the presence of mineralized cartilage flaps or joint mice, and the dimensions of osteophytes in millimeters at the caudal, cranial and medial edges of the joint. Other authors also reported extension of the articular space, joint effusion, swelling of the soft tissues around the joint, fragmentation of medial malleolus or fibular lateral malleolus, radiolucent areas in trochlear ridges, the formation of enthesiophytes and sclerosis of subchondral bone [10,11,18,19,25,30,31].

Mediolateral and plantarodorsal projections, which are the most common, have limited value in the detection of OCD lesions due to the superimposition of trochlear ridges, calcaneus and distal tibia and the fibula [10,28]. Fully flexed and fully extended mediolateral views can be used to partially determine the extent to which the trochlea tali is affected by disease and changes in the size of the articular space between the tibia and the lateral trochlear ridge can be seen [10]. Plantarodorsal views allow for the detection of fewer than half of the lateral ridge lesions [10,11], but according to some authors, it is possible to visualize most of the changes in the medial ridge using this view [17]. Generally, oblique views may help to eliminate superimposition effects [15,17,18,32]. Carlisle et al. [10] stated that the oblique plantaromedial–dorsolateral view with the beam angled at 30–45 degrees is the best for OCD diagnostics. Other authors maintain that the best projection is the fully flexed plantarodorsal (skyline) view, as well as the oblique plantarolateral–dorsomedial view [29].

Computed tomography may be more sensitive to imaging some OCD lesions than radiography, because it eliminates the superimposition of bones and allows for the creation of multiplanar image reconstructions [33,34]. It enables a more accurate assessment of the position, size and number of OCD fragments [18,29,34], as well as an identification of more subtle degenerative changes in the tarsus and better imaging of subchondral bone defects. In over 75% of cases described by Gielen et al. [18], more than two OCD fragments are visible in computed tomography, compared with conventional radiography. However, computed tomography is not the optimal method for imaging soft tissues, including cartilage, because only mineralized cartilage flaps inside the joint may be visible [35]. Other authors do not mention the presence of any hyperdense bony elements inside the affected joint [22]. Dingemanse et al. [36] suggest that increased bone mineral density, which can be measured by computed tomography, may be both a cause and an effect of OCD. Some studies have shown that the areas with the highest density of subchondral bone are also the sites with the highest incidence of OCD lesions [13,36].

Ultrasonographic features of OCD lesions in other joints include joint effusion, hyperechoic joint mice, thickening of the joint capsule, and delineated, irregular areas of cartilage [14]. There is one report in the current literature that described the use of an ultrasound to evaluate the trochlear ridges of the talus [37]. According to Liuti et al. [37], more than 75% of the total area of both ridges can be visualized, meaning that ultrasonography may increase the effectiveness of OCD diagnosis.

## 3. Achilles Tendon Injuries

The common calcaneal tendon (Achilles tendon) is a complex structure, comprising the tendons of the gastrocnemius, a superficial digital flexor and biceps femoris, gracilis and semitendinosus muscles [38,39]. In veterinary medicine, the Achilles tendon, more frequently defined as the common calcaneal tendon, is composed of three tendons that primarily maintain extension of the hock joint, where the gastrocnemius muscle tendon is the largest and most powerful extensor. Injuries to the Achilles tendon are rarely seen in dogs. The most common cause of this pathology is a strong impact which acts in its vicinity and leads to direct ruptures and lacerations [40,41,42]. Most Achilles mechanism disruptions are reported to occur in medium- and large-breed dogs, either during normal walking or as a result of trauma injuries [42]. Some authors consider that the influence of other factors, such as diabetes, obesity, hyperadrenocorticism and the administration of fluoroquinolones, non-steroid anti-inflammatory drugs and corticosteroids, can create conditions under which the Achilles tendon might tear more easily [38,42]. Depending upon the duration and degree of the injury, animals commonly present with an initial non-weight-bearing lameness, which resolves over time. The dogs gradually develop a plantigrade stance, as the talocrural joint becomes more hyperflexed [42].

Radiography can be used for the indirect imaging of Achilles tendon injuries, but its usefulness is limited due to the superimposition of bones and poor contrast, as well as the structure visibility of soft tissues [1,40]. Radiographs can only show the foci of tendon mineralization and the swelling of adjacent soft tissue (Figure 1) [40,43]. Computed tomography (CT) is a cross-sectional imaging modality and has improved the identification of pathological lesions of the calcaneal tendon, such as enthesopathies and tendinopathies. CT provides sectional images; therefore, it eliminates the problems of superimposition correlated with conventional radiology. Achilles enthesopathy is defined as abnormality of the tendons and their attachment to the calcaneal tuber (Figure 2 and Figure 3).

Ultrasonography is a frequently used imaging technique for evaluation of the Achilles tendon and the diagnosis of its injury [14,40,43,44]. Normal anatomy of the common calcaneal tendon has been described elsewhere [14,40,43,44,45]. The Achilles tendon is easily accessible for ultrasound examination, and its superficial parts are clearly visible [14]. According to some authors, it is possible to distinguish individual components of the tendon on the basis of their anatomical relationships and the course of their fibers in the transverse and longitudinal planes [40,43], although Rivers et al. [46] argue that the tendinous structures that build the common calcaneal tendon are not visible.

In the extant literature, there are some descriptions of ultrasound findings of total and partial Achilles tendon ruptures. The best visibility of the tendon can be achieved in the caudal–plantar acoustic window, in the longitudinal and transverse planes [43]. The complete rupture of a tendon appears as a break in the continuity of its fibrillar echostructure (Figure 4). The ends of a tendon are described as “drumstick-like” non-homogeneous structures with increased echogenicity, and they move relative to each other in a dynamic examination. Between them, there is often a hematoma, which appears as a heterogenous, irregularly delineated, more or less anechoic area, with hyperechoic particles inside of it [14,40]. If the rupture is close to the tendon attachment on the calcaneus, its surface is clearly visible. It is possible to visualize an avulsion fracture only if the separated bone fragment is larger than 3 mm [14]. In the case of a partial rupture of the Achilles tendon, the intact part shows its typical appearance, with its fibrillar structure partially preserved. Damaged parts of the tendon lose their normal echogenicity and this area is heterogeneous, hypoechoic-to-anechoic, and irregularly delineated. Sometimes, a thin layer of anechoic fluid can be seen between the tendon and its sheath [14,40]. According to some authors, it is impossible to distinguish between a complete and partial rupture of a tendon [46]. Based on changes in the common calcaneal tendon’s size and echogenicity, ultrasonography may also be useful in monitoring its healing process. However, an ultrasound is unable to determine the exact age of the injury and the time until it might be healed [40].

Magnetic resonance imaging allows for the visualization of tendons and adjacent soft tissue structures in high resolution, multiplanar images (Figure 5). A case report by Lin et al. [47] is the first and only description of the use of MRI in the diagnosis of an Achilles tendon injury in a dog. The tissue at the site of the tendon tear is homogeneous and hyperintense in T2-weighted images, which is similar to previously reported findings in human medicine [48,49,50,51]. The authors also observed the thinning and displacement of tendon fragments and foci of post-contrast enhancement at the site of the injury [47].

Surgical treatment is generally recommended for veterinary patients, in contrast to human medicine, where Achilles injuries are often managed conservatively. Numerous surgical techniques have been described, most involving primary suture repair with secondary immobilization of the talocrural join via transarticular external skeletal fixation, external coaptation, or calcaneotibial positioning screw insertion [42].

## 4. Fractures of the Tarsal Bones

Fractures involving the tarsus are uncommon in dogs. Central bone fractures occur mostly in racing Greyhounds as a result of specific, repeated overloads and maladaption of the medial part of the tarsal joint of the outside racing limb. These are the most frequently reported bone injuries in this population, affecting 64% of Greyhounds [4,5,35]. The right and left hindlimb are unevenly loaded during training and racing on an oval track while the dog runs counterclockwise [3]. This results in a fatigue fracture of the central tarsal bone, with coexisting secondary fractures of the calcaneus [52], talus and tarsal bones from the second to fifth [4,5,53,54,55,56]. Fractures of the central tarsal bone are rarely reported in other dog breeds, such as Border Collies [57,58], Dalmatians [59] or Australian sheepdogs [60]. Isolated fractures of other tarsal bones in dogs are extremely rare. In the current literature, there are single descriptions of these conditions regarding the talus [52,55].

Radiographic examination in the event of a suspected tarsal fracture is similar to that which is performed during OCD diagnosis. The use of this imaging modality is difficult due to the complexity of the tarsus [2]. The overlapping of anatomical structures results in the formation of radiolucent lines that can be misinterpreted as a fracture and complicate the recognition of true lesions [54]. Moreover, small injuries may not be detected if the X-ray beam does not pass parallel to the fracture plane [2]. Many authors have used more than two orthogonal views (mediolateral and plantarodorsal) to accurately assess the tarsus [2,3,28,29,52,61], including stress projections [10,18,62,63]. According to Guilliard [58], some fractures of the central tarsal bone are visible in two basic views. Butler et al. [2] found that a 10-view study was comparable to a 2-view study when evaluated by an experienced radiologist. The use of 10 views in a dog after trauma is less feasible in clinical practice [34]. The time needed to perform 10 radiographs with the necessity of constantly repositioning the joint can increase the pain and anxiety of the patient, and potentially increase joint damage [2].

Computed tomography is better than radiography in detecting small bone fragments, and it is more reliable in imaging displacements within the fracture [34,54]. It often allows for a correct diagnosis and the planning of surgical treatment. Butler et al. [2] found that the sensitivity of computed tomography was subjectively higher (77% versus 57%, respectively) compared to radiographic examination in ten projections. According to the authors, the sensitivity of fracture detection was subjectively higher for each bone in computed tomography, and small chipping fractures of the central tarsal bone were detectable only with this imaging technique. Nevertheless, false positive results were noted more frequently in CT examination than in radiography [2]. These results are consistent with those obtained by Hercock et al. [54]. According to Hercock et al., the use of computed tomography for fracture evaluation and classification improved the observer’s ability to correctly assess fractures [54].

Similar to the evaluation of OCD lesions, a bone mineral density assessment may be used; these findings could potentially help to minimize the risk of central bone fractures. Bergh et al. [3] found that dogs with tarsal fractures had higher bone density than those without lesions. Furthermore, there are no differences in the total bone mineral density between the affected and healthy limb [3]. According to Thompson et al. [56], a computed tomographic study of bone modeling is needed to track skeletal adaptations in racing Greyhounds, especially after they have begun training and racing in a counterclockwise fashion.

## 5. Conclusions

Radiography is a relatively inexpensive and readily available imaging technique that, as a rule, requires sedation, although not always general anesthesia [1]. It is useful primarily in the diagnosis of fractures and osteochondrosis. The complexity of the tarsal joint may make it difficult to interpret radiographs because structures overlap and only generalized soft tissue swelling is visible [1,2,10,34,54]. According to Hercock et al. [54], the evaluation of fractures exclusively by radiography results in many fractures being misclassified as less severe. Both in the diagnosis of fractures and OCD, it is necessary to obtain more than two views, including oblique projections. Despite this, according to some authors, radiography shows only a 78% sensitivity in the diagnosis of some types of tarsal OCD [29].

An ultrasound of muscles and tendons is more accurate than radiography in the evaluation of soft tissues. It is easily accessible and does not require general anesthesia. It may also be an additional method for bone evaluation. However, in the case of a fracture, only the contours of bone discontinuity are visible [14]. In veterinary medicine, ultrasound is the primary imaging method for assessing tendinopathy and it should be used to locate tendon injuries and determine the severity of their damage. It also allows for postoperative control, for monitoring the healing of a torn tendon, and for the assessment of a presented condition which might suggest an appropriate recovery time [14,40]. The visibility of the individual components of the Achilles tendon varies between reports by different authors [40,43,46]. This can be explained by the frequency and type of the transducer, as well as by the examination technique. According to Kramer et al. [14], most of the soft tissue structures of the tarsus can be visualized by an ultrasound in dogs weighing more than 15 kg; in smaller ones, it is more difficult to visualize small structures. Osteochondral defects may be difficult to observe in the tarsal joint due to its narrow articular spaces and the small acoustic window.

According to many authors, computed tomography is the method of choice in the diagnosis and assessment of pathological changes in the ankle joint [34,64,65]. On a CT scan, the structures do not overlap, as in classical radiography. By using appropriate imaging windows of various greyscales, it is possible to perform a detailed evaluation of bone structures. It has limited use in the diagnosis of changes in soft tissues, but the use of a contrast agent is possible. CT also allows for the multiplanar reconstruction of images [33,34]. The above-mentioned advantages of this imaging method facilitate the assessment of complex fractures, as well as OCD lesions. CT also enables the detection of subchondral sclerosis by measuring the difference in density between the cortical and spongy bone, and of changes in the thickness of the cortical bone [34]. CT has been shown to be superior to radiology in assessing and classifying severe fractures of the central tarsal bone and improving the observer’s ability to correctly identify the majority of adjacent tarsal fractures. However, CT does not detect all pathological features, and fractures of the smallest bones may still be overlooked [54].

Compared to other imaging methods, magnetic resonance imaging provides the best contrast of soft tissues. Its increasing use has been observed, but the cost of an examination and the need to perform it under general anesthesia are limiting factors. High resolution, multiplanar images facilitate the assessment of both soft tissues and bones, and enable the detection of disease processes involving muscles, tendons and ligaments. MRI may also have potential applications in the diagnosis of tendinopathy and partial or complete tears of tendons and ligaments. It allows the visualization of the tendons and all surrounding structures in one image, which enables the assessment of anatomical relationships between them. It can complement an ultrasound, allowing for the accurate high-resolution imaging of a tendon injury [47]. In the case described by Lin et al. [47], changes in the signal, indicating swelling and alterations in the thickness of the Achilles tendon, were easily visible. The use of a contrast agent facilitates identification and enables the confirmation of the presence of a neoplastic process or inflammation within the joint [47]. MRI has the potential to replace CT in the diagnosis of OCD. This is the technique of choice in the diagnosis of OCD in humans. It enables an accurate evaluation of cartilage and subchondral bone [66]. Canine articular cartilage is thin; therefore, an assessment of the cartilage in the narrow articular gaps in the tarsus may be difficult [67]. In summary, all the techniques described above are used in the diagnosis of lesions in the canine tarsal joint, but only the simultaneous use of several imaging methods, in conjunction with clinical examination, allows for a full diagnosis.

## Figures and Tables

**Figure 1 animals-11-01834-f001:**
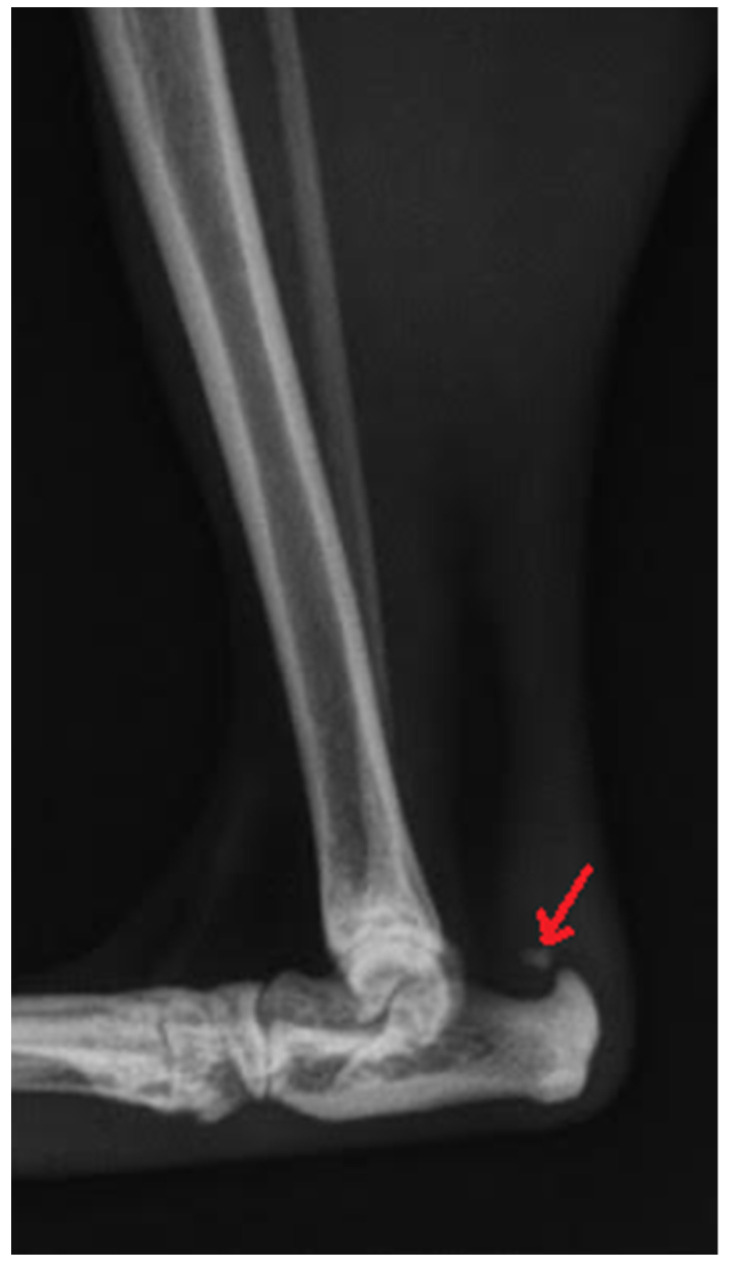
Radiographic image of total Achilles tendon rupture. The focus of tendon mineralization (arrow) is shown.

**Figure 2 animals-11-01834-f002:**
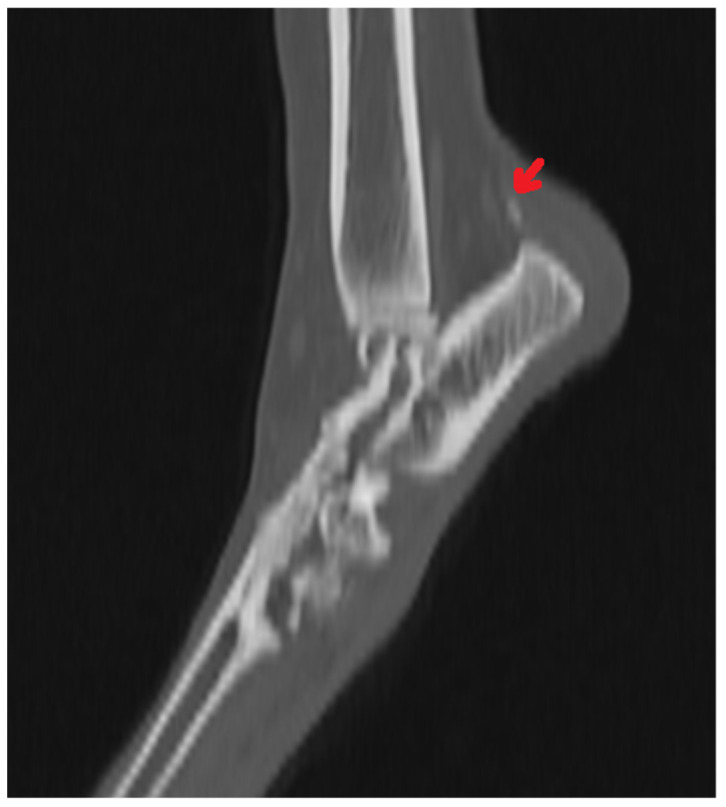
CT image of the mineralization of the Achilles tendon (arrow).

**Figure 3 animals-11-01834-f003:**
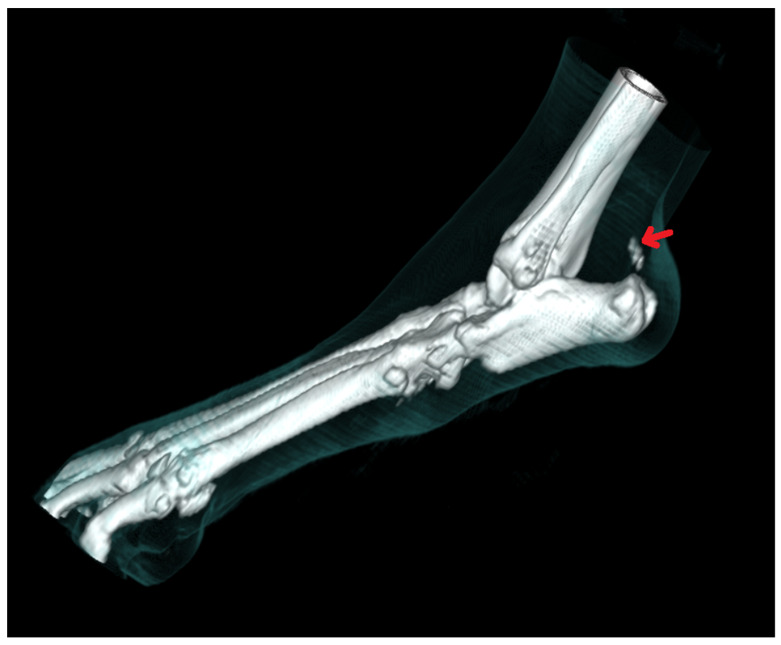
CT 3D image reconstruction of the mineralization of the Achilles tendon (arrow).

**Figure 4 animals-11-01834-f004:**
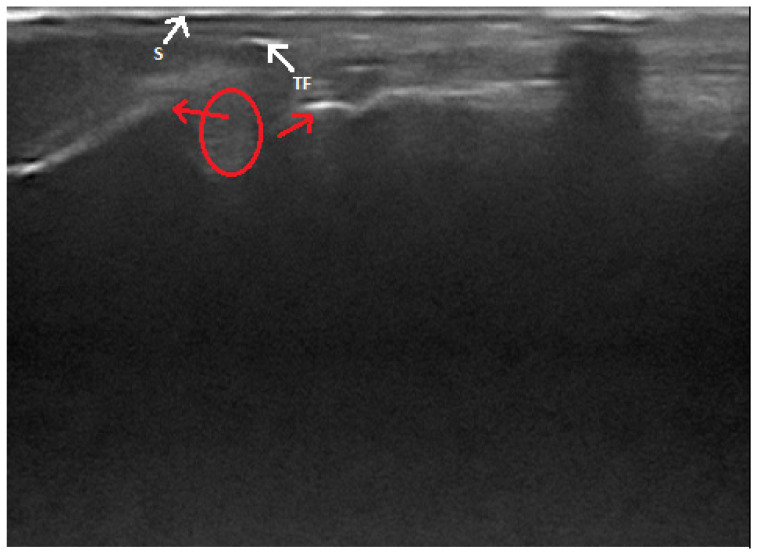
Ultrasonographic image of total Achilles tendon rupture. Ends of the torn tendon (red arrows), a hematoma (circle), a fragment of a tendon (TF) and the skin surface (S) is shown.

**Figure 5 animals-11-01834-f005:**
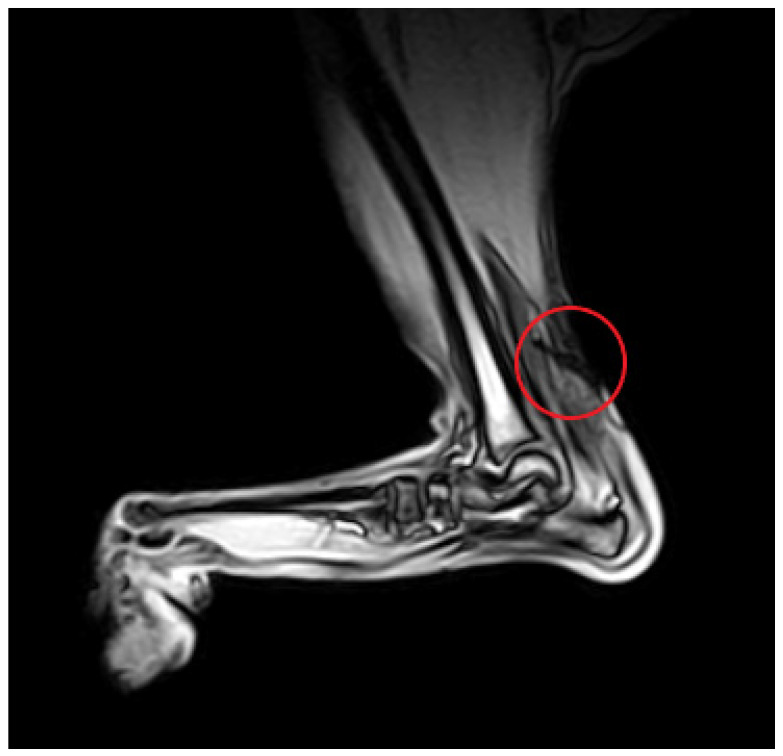
MRI GE T1-weighted image of a partial Achilles tendon rupture (circle).

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
