# Peer review of "Usefulness of Imaging Techniques in the Diagnosis of Selected Injuries and Lesions of the Canine Tarsus. A Review"

_animals, 2021, doi:10.3390/ani11061834_

Round 1

Reviewer 1 Report

This is a good novel paper which compares the different available imaging modalities in their usefulness to diagnose tarsal injuries. It appears to be a review of the literature which should be explained to the reader in the title and abstract. The paper is difficult to read as a result of the large number of grammatical errors and  requires a good check for grammar as there are many errors.

Each section lists a description of the condition and the way each modality diagnoses that condition. This needs to be written more clearly in each section.      

Line 21 – these conditions instead of this conditions

Line 24 – Setence beginning aim of this paper  doesn’t quite make sense, can you reword for clarity.

Line 33 – Lack of support not lacks

Line 43. – In the joint

Line 44 – relatively rarely

Line 57 – Sentence beginning The cost is decreasing – do you equipment is becoming more widely available?

Line 60  - affecting

Line  62 – in the literature

Line 64 – approximately

Line 67 – of overall cases

Line 69 – sentence beginning medial trochlear ridge doesn’t make sense. Do you mean the literature reports?

The second paragraph under osteochondritis beginning lesions characteristic could benefit from a re-write to fix grammatical errors and make it more succinct.

Line 83 – Sentence beginning in addition – do you mean taking multiple views?

Line 89 – is presence and size a criteria for diagnosis? I’m a bit confused by what this paragraph is trying to say sorry. Is this to diagnose them or is this the information recommended to be recorded to assess severity?

Line 114 – What does the sentence beginning In over 75% of cases mean? It doesn’t make sense sorry. I’m not sure what you are trying to say here.               

Line 117 – what do you mean by the sentence another studies do not mention….

Line 120 – Other studies?

Line 122- include not includes

Line 124 – in the current literature

The osteochondritis section has some interesting information however it could be re-written to be more succinct.

Line 129 – consisting.

Line 132 – pathology

Line 134 – such as.

Line 133 – the sentence beginning some authors – doesn’t make sense. Do you mean some authors suggest that the list things increase the tendency of the Achilles to tear?
Line 171 – The size before visualisation of an avulsion fracture will depend on the ultrasound system being used. Is there a common system used in canine’s, if not you should give the system used.

Line 192 – Are you referring to greyhounds in this sentence? It is a little confusing.

Line 197 – should border collie be capitalised like all the other dog breeds?

Line 198-  What do you mean by single descriptions?

Line 210 – by patient do you mean the dog?

There is no mention of imaging using ultrasound for tarsal bones. Ultrasound isn’t usually used for bone fractures but is able to see soft tissue adjacent reaction.

Conclusion – the operator experience and the type of ultrasound machine are also key factors in determining whether ultrasound can assess individual tendon components. MRI is only mentioned in the conclusion. It would be good to mention this on the way through as you have with the other modalities. As you have not reported any literature on MRI in canines it might be prudent to leave this out. It would also be good to present images from each modality throughout.

The paper would be beneficial for those looking at different imaging modalities in imaging canine tarsal injuries. I would encourage the authors to do a full re-write to improve the grammar and the clarity so that the paper clearly presents the advantages and disadvantages of each modality for each injury. If there are no papers on MRI, it may be beneficial to leave this out. How often is MRI used for canine imaging?

Author Response

Dear Reviewer,

thank you for your thorough analysis of our Manuscript and for constructive comments. In the attachment I am sending response to the comments.

Yours sincerely,

Justyna Abako

Reviewer 2 Report

The paper is well written and describes the orthopaedic conditions of the canine tarsal joint and the appropriate approach to diagnosis.  It does need some grammatical changes and would benefit from a review by an expert in English. 

One error that needs correction is to replace 'superficial digital extensor' with 'superficial digital flexor' on line 130. 

Author Response

(The authors gave the same response as above.)

Reviewer 3 Report

General comments

The paper submitted by Abako et al. characterize and compare the usefulness of imaging techniques available in veterinary medicine for the diagnosis and evaluation of lesions and injuries which affect the tarsal joint in dogs. This manuscript gives adequate information about different processes. Nonetheless, there are important recommendations that must be taken into consideration to improve the manuscript quality. Therefore, the length of the manuscript is inappropriate to be considered as a Review. Similar mistakes are shown in the reference list, where only 10% of the papers cited are within the last five years.

As the authors explain, radiography is valid, but its sensitivity is lower than newer imaging techniques, such as CT or MRI. However, there are no CT or MRI figures in the manuscript, only x-rays.

Specific comments

The authors must describe the sensitivity found with the different diagnostic techniques.

Line 114: “In over 75% of cases more than two fragments are visible in computed tomography than in radiography”, this sentence is confusing; please explain.

It would be interesting to compare CT results in other joints as you do with ultrasonography.

The information provided in figure 1 is not well visualized. Inflammation of adjacent soft tissue structures is not so precise as it is stated. In addition, the radiological findings are not labeled. Please also check the picture resolution.

Line 160-161, new ultrasonography equipment and linear probes give a great definition. Please check

Fracture of the tarsal bones

Please add arrows in both views to better visualize the fractures. Moreover, it seems that the fracture affects the distal part of the tibia, no the tarsal area.

Line 230, please explain this sentence better since it can be misinterpreted.

Author Response

(The authors gave the same response as above.)

Round 2

Reviewer 1 Report

The paper is very much improved. It still requires extensive editing for grammar and is difficult to understand in this respect. The content is good, however the writing style makes it quite difficult to read. 

There are many errors I did not point out and there is no response to the broader comment that the above needed to be addressed. 

Author Response

Dear Reviewer,

Thank you for your thorough analysis of our Manuscript and for constructive comments. I corrected the article according to the instructions of the MDPI English editing.

Best regards,

Justyna Abako

Reviewer 3 Report

In this revised version, there are some essential details must be provided before acceptance. Therefore, the English style should be carefully revised.

In addition, the authors have included ultrasound and MRI images. However, in the ultrasound figure, some details  are not labeled. Moreover, the authors must include other views in both figures since they do not provide CT figures.

In figure 1, remove "the area of swelling" since it is not seen

Please, crop the data showed in the ultrasound figure.

Author Response

Dear Reviewer,

thank you for your thorough analysis of our Manuscript and for constructive comments. I have corrected the figures according to your instructions.

Best regards,

Justyna Abako